# Pharmacological Inhibition of Lipid Import and Transport Proteins in Ovarian Cancer

**DOI:** 10.3390/cancers14236004

**Published:** 2022-12-05

**Authors:** Lisa Lemberger, Renate Wagner, Gerwin Heller, Dietmar Pils, Thomas W. Grunt

**Affiliations:** 1Cell Signaling and Metabolism Networks Program, Division of Oncology, Department of Medicine I, Medical University of Vienna, 1090 Vienna, Austria; 2Comprehensive Cancer Center, Medical University of Vienna, 1090 Vienna, Austria; 3Division of Oncology, Department of Medicine I, Medical University of Vienna, 1090 Vienna, Austria; 4Division of Visceral Surgery, Department of General Surgery, Medical University of Vienna, 1090 Vienna, Austria; 5Ludwig Boltzmann Institute for Hematology and Oncology, 1090 Vienna, Austria

**Keywords:** fatty acid, import, inhibitor, lipid, lipid handling protein, molecular targeting, ovarian cancer, transport, uptake

## Abstract

**Simple Summary:**

Ovarian cancer (OC) is still the most lethal gynecological cancer due to late diagnosis when peritoneal metastasis has already occurred. OC progression requires lipid nutrients that are either synthesized endogenously in the cancer cells or imported from the surrounding host tissue. Accordingly, blockade of lipid synthesis has been shown to be a powerful strategy against OC. However, direct evidence of the role of lipid import and transport for OC cell growth is still largely missing. Therefore, we exposed OC cells to inhibitors of lipid uptake and transport proteins, which are typically overexpressed in OC. Our data reveal that pharmacological inhibition of these lipid handling proteins caused a drug-specific, dose-/time-dependent decline of lipid uptake, which was associated with cell growth reduction, cell cycle arrest, and apoptosis, indicating that OC cells are exquisitely sensitive to lipid deficiency. This dependency provides the rationale for the development of novel lipid-antimetabolic strategies against OC.

**Abstract:**

Ovarian cancer (OC) is the most lethal gynecological malignancy with a 5-year survival rate of 49%. This is caused by late diagnosis when cells have already metastasized into the peritoneal cavity and to the omentum. OC progression is dependent on the availability of high-energy lipids/fatty acids (FA) provided by endogenous de novo biosynthesis and/or through import from the microenvironment. The blockade of these processes may thus represent powerful strategies against OC. While this has already been shown for inhibition of FA/lipid biosynthesis, evidence of the role of FA/lipid import/transport is still sparse. Therefore, we treated A2780 and SKOV3 OC cells with inhibitors of the lipid uptake proteins fatty acid translocase/cluster of differentiation 36 (FAT/CD36) and low-density lipoprotein (LDL) receptor (LDLR), as well as intracellular lipid transporters of the fatty acid-binding protein (FABP) family, fatty acid transport protein-2 (FATP2/SLC27A2), and ADP-ribosylation factor 6 (ARF6), which are overexpressed in OC. Proliferation was determined by formazan dye labeling/photometry and cell counting. Cell cycle analysis was performed by propidium iodide (PI) staining, and apoptosis was examined by annexin V/PI and active caspase 3 labeling and flow cytometry. RNA-seq data revealed altered stress and metabolism pathways. Overall, the small molecule inhibitors of lipid handling proteins BMS309403, HTS01037, NAV2729, SB-FI-26, and sulfosuccinimidyl oleate (SSO) caused a drug-specific, dose-/time-dependent inhibition of FA/LDL uptake, associated with reduced proliferation, cell cycle arrest, and apoptosis. Our findings indicate that OC cells are very sensitive to lipid deficiency. This dependency should be exploited for development of novel strategies against OC.

## 1. Introduction

Ovarian cancer (OC) represents the seventh most common form of cancer in women (225,500 cases per year) and the eighth leading cause of cancer lethality worldwide (140,200 deaths per year) [1,2]. This unfavorable outcome is mainly due to late diagnosis, when the cells of the primary tumor have already undergone epithelial–mesenchymal transition. During this process, the intercellular contacts become loose, allowing for the detachment of the cancer cells from the basement membranes and exfoliation into the abdominal cavity, where they are freely floating in the peritoneal fluid giving rise to malignant ascites. The metastatic ascites cells preferentially colonize the omental fat pad, an adipocyte-rich tissue located behind the visceral peritoneum [1,3,4]. The metabolism of cancer cells differs in many aspects from that of normal cells and gives rise to a characteristic malignant metabolic phenotype, which has recently been recognized as another crucial hallmark of cancer [5]. Both microenvironments, the ascitic as well as the omental, provide an abundance of energy dense lipids and fatty acids (FAs) that trigger metabolic reprogramming of the cancerous cells by shifting from glycolysis to lipid metabolism and spur metastatic progression, spreading, invasion, and homing of the malignant cells. The high metabolic plasticity of OC cells and the particular growth conditions enable the cells to use the complex mixture of bioactive lipids and free FAs, obtained from the ascites fluid and from omental adipocytes, for multiplication, growth, and survival during abdominal metastasis [4,5,6,7,8,9,10]. In addition, OC cells can autonomously produce FAs through a process known as endogenous de novo fatty acid synthesis [11,12,13,14,15,16,17,18]. Imported and endogenous lipids serve together as sources for signaling molecules, as building blocks for new membranes and as carriers of energy. Overall, there is increasing evidence that OC cells are exquisitely dependent on lipids for growth, proliferation, progression, and metastasis. Accordingly, we and others have previously shown that blockade of FA synthesis effectively abrogates OC cell growth and metastasis [11,12,13,16,18]. Thus, targeting endogenous FA synthesis holds promise for OC treatment.

Here, we take this lipid-antimetabolic approach to the treatment of OC further and examined a related experimental strategy that would complement the antilipogenic procedure. Accordingly, we hypothesize that disrupting the import and transport of exogenous FAs and lipids in the malignant cells represents another new and promising way to block the growth of OC. Support for this notion comes from the fact that the lipid uptake proteins’ fatty acid translocase/cluster of differentiation 36 (FAT/CD36) and low-density lipoprotein (LDL) receptor (LDLR), and the intracellular lipid transporters of the fatty acid-binding protein (FABP) family, as well as fatty acid transport protein-2 (FATP2 aka SLC27A2) and ADP-ribosylation factor 6 (ARF6), are overexpressed in OC and other malignancies and correlate with aggressive behavior [14,19,20]. These lipid chaperones support cell growth, proliferation, metastasis, and progression by maintaining cellular membrane FA levels and facilitating lipid transport across the endoplasmic reticulum, Golgi, mitochondria, peroxisomes, lipid droplets, and nucleus [21,22,23,24]. Although their crucial role in cancer cell biology has already been well documented, their utility as sensitive drug targets still remains largely unexplored. In OC in particular, there is only one recent report demonstrating the anticancer efficacy of the FABP4 inhibitor BMS309403 [25]. Therefore, our aim was to directly interfere with lipid import and transport pathways in OC cells under well-defined conditions in culture by inhibiting FAT/CD36, FABPs, FATP2/SLC27A2, or ARF6. Here, we show that exposure of OC cell cultures to small molecule inhibitors of these lipid handling proteins caused drug-specific, dose-/time-dependent inhibition of FA and LDL uptake. This was associated with growth reduction, cell cycle arrest, and apoptosis. Our data confirm the assumption that OC cells are highly dependent on adequate lipid supply. This addiction creates a potential Achilles heel that could be directly and effectively targeted during the treatment of OC. This new approach should be examined in subsequent in vivo studies.

## 2. Material and Methods

### 2.1. Cells, Culture Conditions, and Reagents

The endometrioid ovarian carcinoma cell line A2780 (M. Krainer, Medical University Vienna, Vienna, Austria) [26], the HGSOC cell line OVCAR3 (ATCC, Manassas, VA, USA), and the SKOV3 cell line (ATCC), which reveals features of both HGSOC and ovarian clear cell carcinoma [27], will be subsumed below under the acronym OC. They were maintained in RPMI1640 or in alpha-MEM (Gibco, Karlsruhe, Germany) [28]. Freshly thawed cells were grown for a few passages in media supplemented with 10% fetal bovine serum (FBS) containing 100 IU (mg)/mL penicillin–streptomycin and 2 mM glutamine (Gibco). Cultures were grown at 37 °C, 5% CO_2_, and 95% humidity, and cells were checked for negativity of viruses/bacteria/fungi/mycoplasma (Venor GeM, Minerva Biolabs, Berlin, Germany). Species’ origin was demonstrated by species PCR, and cell line identity was tested by fluorescent nonaplex PCR of short tandem repeat markers (DSMZ, Braunschweig, Germany). Once cell cultures were established, the concentration of FBS was reduced to 5%, which is sufficient for support of stable growth. Natural heparin sodium salt from porcine intestinal mucosa was purchased from Sigma-Aldrich (St. Louis, MO, USA), reconstituted in sterile distilled water, and diluted in media. The names, target specificities, suppliers, and references of all synthetic small molecule inhibitors used in this study are summarized in Table 1. Lyophilized drugs were dissolved in 100% DMSO, and the final dilution was 1:1000 in media.

### 2.2. Western Blotting

OC cells (20,000/cm^2^ in 35 mm or 60 mm dishes, Corning, Corning, NY, USA) were allowed to adhere overnight and were cultured for 48 or 72 h in media with 5% FBS ± inhibitor. After lysis, cellular proteins were subjected to SDS–PAGE, blotted, and immunostained as previously described [35,36] using anti-FABP4 (1:200, #2120, Cell Signaling Technology, Boston, MA, USA), anti-FABP5 (1:500, #39926, Cell Signaling Technology), anti-CD36 (1:1000, #14347, Cell Signaling Technology), anti-ARF6 (1:500, #3546, Cell Signaling Technology), anti-FABP6 (1:250, #96122, Abcam, Cambridge, UK), anti-LDLR (1: 1000, #52818, Abcam), anti-FATP2 (1:500, # 175373, Abcam), and anti-actin (1:1000, #1616, Santa Cruz Biotechnology, Dallas, TX, USA). Secondary antibodies were peroxidase-labeled donkey anti-rabbit (1:15,000, #16284, Abcam), donkey anti-goat IgG (1:15,000, #2020, Santa Cruz Biotechnology), donkey anti-mouse (1:10,000, #715-035-150, Jackson ImmunoResearch, West Grove, PA, USA), or donkey anti-sheep (1:10,000, #A16041, Thermo Fisher, Waltham, MA, USA). Detection was by enhanced chemiluminescence.

### 2.3. Cell Proliferation

OC cells were seeded at a density of 23,500/cm^2^ in 96-well plates and were allowed to adhere overnight in medium containing 5% FBS before various concentrations of small molecule inhibitors were added. After indicated periods of time, viable cell numbers were determined with a formazan dye assay (#BI-5000, Biomedica, Vienna, Austria) as described [37]. This assay is essentially an improved MTT assay, which uses a non-toxic tetrazolium salt that is reduced by mitochondrial dehydrogenases to a colored formazan product. Under the prevailing cell culture conditions, there is a linear relationship between the optical density values obtained and the number of viable cells; therefore, the results are reported as ‘Cell Numbers’ for convenience. In the initial experiments, the results obtained by formazan dye assay were compared with direct hemocytometer counting.

### 2.4. Apoptosis Assays

To discriminate between early and late stages of drug-induced apoptosis, two different fluorescent labeling techniques were used. The highly fluorescent allophycocyanin (APC) annexin V conjugate (Biolegend, San Diego, CA, USA) detects externalization of phosphatidylserine at the cell membranes—a process that starts very early during active apoptosis—whereas positivity for the fluorescent DNA intercalating agent propidium iodide (PI) (Sigma-Aldrich) indicates loss of membrane integrity/increase of cell permeability, which occurs during late stages of apoptosis or passive necrosis. Intermediate stages of apoptosis were examined in another set of experiments by incubation of the cells with a phycoerythrin (PE)-labeled anti-active caspase-3 antibody (#561011, Becton Dickinson, Franklin Lakes, NJ, USA). Activated caspase-3 acts as an executioner caspase. It coordinates the cellular self-destruction process by inducing DNA fragmentation and degradation of the cytoskeleton. All procedures were performed according to previous published protocols [38,39,40]. Briefly, treated and control cells seeded at a density of 36,000/cm^2^ in T25 cell culture flasks were harvested; rinsed with PBS; exposed to APC annexin V diluted in binding buffer containing HEPES (10 mM, pH 7.4), NaCl (140 mM), and CaCl2 (2.5 mM); rinsed again and labeled with PI (1 mg/mL). For detection of active caspase-3-positive cells, samples were fixed with formaldehyde (2%), permeabilized in methanol (100%) for 15 min at –20 °C, and stained with PE-labeled anti-active caspase-3. Labeled cells were then analyzed in a FACSCalibur flow cytometer (Becton Dickinson).

### 2.5. Cell Cycle Analyses

Untreated and treated cells grown at a density of 36,000/cm^2^ in T25 cell culture flasks were harvested, rinsed in PBS, and fixed with 70% ethanol for 15 min at 4 °C before incubation in 0.05% Triton X-100/PBS, RNase A (100 μg/mL), and PI (50 μg/mL) for 40 min at 37 °C. Cells were finally rinsed in PBS and analyzed in a FACSCalibur. Received data were analyzed with ModFit software (Verity Software House, Topsham, ME, USA) as previously described [38,39].

### 2.6. FA and LDL Uptake Assays

OC cells seeded at a density of 23,500/cm^2^ in black 96-well plates with an optical bottom (Nunc, Roskilde, Denmark) were allowed to adhere overnight and then exposed for 0–48 h to the indicated concentrations of small molecule inhibitors of cellular lipid handling proteins in media containing 5% FBS before drug-mediated time-/dose-dependent alterations in FA and LDL uptake rates were ascertained. Drug concentrations that caused at least 50–60% changes of FA or LDL uptake after 48 h of exposure were used for subsequent time-course analyses after 1, 2, 4, 24, and 48 h of drug treatment. Cells were subsequently extensively rinsed and processed for labeling with a BODIPY-dodecanoic acid fluorescent fatty acid analog (BODIPY 500/512 C1, C12; contained in the QBT™ Fatty Acid Uptake Kit from Molecular Devices, San Jose, CA, USA) or with the pH-sensitive fluorochrome pHrodo™ Red conjugated to LDL (pHrodo™ Red LDL Uptake Kit from Thermo Fisher). After incubation for approximately 10 min (QBT™ Fatty Acid Uptake Kit) or 4 h (pHrodo™ Red LDL Uptake Kit), the fluorescent signals were determined following the manufacturers’ protocols and detected in a SpectraMax iD3 multi-mode microplate reader (Molecular Devices). Obtained signals were normalized to cell numbers that were concurrently determined in a formazan dye assay and in a hemocytometer. Solvent (0.1% DMSO)-treated control cells cultured in medium with 5% FBS were arbitrarily set to 100%, and experimental data were then expressed in a percent relative to these controls (for further details, see [14]).

### 2.7. NADP and NADPH Assay

Whole-cell lysates were used to determine the content of NADP and NADPH in cells grown in media containing solvent with or without inhibitors of lipid handling proteins. Briefly, A2780 cells were cultured as described for FA and LDL uptake assays, followed by 48 h of drug treatment. After cell lysis, the amounts of NADP and NADPH were measured using an NADP/NADPH fluorometric assay following the manufacturer’s (Abcam) instructions.

### 2.8. RNA-Sequencing

A2780 and SKOV3 OC cells were plated at a density of 36,000/cm^2^ in T75 cell culture flasks and treated for 4 h or 48 h with solvent or with small molecule inhibitors of lipid handling proteins before being harvested. Total RNA was isolated from cell pellets using Qiagen’s (Hilden, Germany) RNeasy kit according to manufacturer´s instructions. RNA-seq libraries were prepared using the QuantSeq 3′ mRNA-Seq FWD library preparation protocol (Lexogen, Vienna, Austria) and sequencing (50 base-pair, single-end) was performed on a HiSeq 3000 instrument (Illumina, San Diego, CA, USA). HiSeq Control Software (HCS, HD 3.4.0.38) and Real-Time Analysis Software (RTA, 2.7.7) were used for raw data acquisition and base calling, respectively. Fastq files were adapter and quality trimmed using Trim galore! (v.0.4.4) followed by alignment to the human genome GRCh38 by STAR (v.2.5.2) [41]. R software (v4.0.4, Nokia Bell Labs, Murray Hill, NJ, USA) and DESeq2 (v.1.30.1, Bioconductor, Fred Hutchinson Cancer Research Center, Seattle, WA, USA) were used to calculate differential gene expression (cut-offs: |shrunkFC| > 1.5 and FDR < 0.05) [42]. Functional enrichment analyses were performed using the R package clusterProfiler (v4.5.1, Bioconductor) [43]. RNA-seq data were deposited at GEO database (GSE212702).

### 2.9. Statistical Analysis

Results are presented as mean values (Mean) ± standard deviation (SD) of at least three independent experiments. Significant differences between control and treated groups were identified by ANOVA and post hoc Scheffe test or by Student’s *t*-test at levels of significance of *p* < 0.05 (*), *p* < 0.01 (**), and *p* < 0.001 (***), as appropriate.

## 3. Results

We have previously shown that growth and lipid metabolism of OC cells is highly dependent on endogenous de novo synthesis of FA, corroborating the findings of others who suggest that FA synthesis pathways provide promising targets for cancer drug development [12,13,44]. Moreover, we recently demonstrated that OC cells contain a variable and ample repertoire of lipid handling proteins, including transmembrane FA- and lipid-receptor proteins, such as CD36 and LDLR; cytoplasmic FA-binding proteins FABP4, FABP5, and FABP6; and the anabolic FA-producing enzymes ATP citrate lyase (ACLY), acetyl-CoA carboxylase(ACC)1, ACC2, and fatty acid synthase (FASN) [14].

### 3.1. The Effects of Inhibitors of Lipid Handling Proteins on the Proliferation and on the Expression of the Targeted Proteins in OC Cells

In this study, we investigated whether proteins that facilitate the uptake of FA and lipids from the tissue microenvironment into the OC cells and mediate their subsequent intracellular transport could represent additional targets for the development of potent anticancer drugs. To this end, OC cells were exposed to various inhibitors of lipid handling proteins, and the effects on cell proliferation and on corresponding target protein expression were examined in the presence or absence of the inhibitors. The names and specific binding partners/targets of the applied inhibitors are given in Table 1. Exposure to these agents induced dose- (Figure 1) and time-dependent (Appendix A) reductions of cell numbers in A2780 (Figure 1A,B), SKOV3 (Figure 1C,D), and OVCAR3 (Figure 1E,F) cells as shown by formazan dye assay, albeit with different efficacies. In A2780, the strongest antiproliferative effects were obtained with NAV2729 (IC_50_ 8.9 µM), followed by HTS01037 (IC_50_ 36.5 µM), SB-FI-26 (IC_50_ 37.0 µM), and BMS309403 (IC_50_ 48.1 µM). The latter three showed very similar activity, whereas sulfosuccinimidyl oleate (SSO; IC_50_ 166.8 µM) was less active (Figure 1B). In SKOV3, similar sensitivities were obtained with NAV2729 (IC_50_ 10.3 µM), HTS01037 (IC_50_ 35.0 µM), and BMS309403 (IC_50_ 71.2 µM), whereas SB-FI-26 (IC_50_ 87.0 µM) and especially SSO (IC_50_ 271.5 µM) were hardly effective (Figure 1D). OVCAR3 cells also revealed sensitivity against NAV2729 (IC_50_ 7.1 µM) and HTS01037 (IC_50_ 26.8 µM), but less to SB-FI-26 (IC_50_ 50.7 µM) and BMS309403 (IC_50_ 60.5 µM) (Figure 1F). Overall, the FATP2/ARF6 inhibitor NAV2729 was the strongest growth inhibitor, and the group of FABP inhibitors (HTS01037, SB-FI-26, BMS309403) revealed less or roughly similar efficacy, while CD36-binding SSO was the least effective. This was seen in cells not only of the endometrioid type (A2780), but also in cells of the high-grade serous ovarian (adeno)carcinoma (HGSOC)-(OVCAR3) or mixed HGSOC/clear cell carcinoma type (SKOV3), with A2780 being the most sensitive and SKOV3 the least sensitive. It was therefore of interest to compare these two cell types in more detail. To this end, we applied Western blotting and examined the effects of these lipid import and transport inhibitors on the expression of their corresponding target proteins in A2780 and SKOV3, but not in OVCAR3. Interestingly, while none of the inhibitors affected expression of the cognate target proteins (FATP2, ARF6, FABPs, CD36), all but SSO downregulated LDLR levels in both cell lines. This decrease in LDLR can be largely attributed to accelerated protein degradation, because smaller protein fragments that were reactive with the anti-LDLR antibody were detected in Western blots of both cell lines upon exposure to some of the inhibitors (Figure 2; the uncropped Western blots are shown in Appendix A), and in most cases, no reduction in corresponding mRNA levels was seen by RNA-seq at the transcriptomic level (except some downregulation of LDLR mRNA in SKOV3 after HTS01037 and SB-FI-26) (Appendix A, and whole RNA-Seq data set deposited in the GEO database at GSE212702). The CD36 inhibitor SSO, on the other hand, caused a slight to moderate downregulation of FABP4, FABP5, FABP6, FATP2, and ARF6 in A2780, and weak reductions in SKOV3 cells, as shown by Western blotting (Figure 2). In summary, SSO caused weak to moderate downregulation of all examined lipid handling proteins at the protein level but not at the mRNA level, except LDLR, which was slightly upregulated in both cell lines. Some drugs (e.g., NAV2729, BMS309403, and HTS01037) altered FATP2 mRNA, but this effect was only seen in SKOV3 cells and was not translated to the protein level. Overall, the data suggest a post-transcriptional mechanism of gene regulation with NAV2729, BMS309403, HTS01037, and SB-FI-26 primarily affecting LDLR levels in both cell lines, while SSO downregulates FABP proteins, FATP2, and/or ARF6 in A2780, as well as SKOV3 to a lesser extent.

### 3.2. Inhibitors of Lipid Handling Proteins Diminish the Uptake of Exogenous Free FA into OC Cells

Next, we investigated the functional on-target effect of the applied lipid uptake and transport inhibitors on the sensitive A2780 and the relatively resistant SKOV3 cells. As expected, fluorescent labeling of lipid-depleted A2780 and SKOV3 cells with BODIPY-tagged fluorescent dodecanoic acid revealed that the inhibitors of lipid handling proteins caused dose- and time-dependent downregulation of free FA uptake. Overall, the dose dependencies for growth inhibition and for reduction in free FA uptake were relatively similar (compare Figure 1A–D with Figure 3A–D). In both cell lines, the FATP2/ARF6 inhibitor NAV2729 was most potent, followed by the FABP inhibitors HTS01037 and BMS309403, while CD36-targeting SSO was much less effective after 48 h of exposure (Figure 3A,C). The resulting IC_50_ values for inhibition of FA uptake are given in Figure 3B,D and were used as guidance for choosing roughly equipotent drug concentrations for subsequent time course experiments. Of note, SKOV3 cells, which proved to be relatively resistant to growth reduction by the FABP5 inhibitor SB-FI-26 (Figure 1C,D), were found to be also resistant to SB-FI-26-mediated blockade of free FA uptake (Figure 3C). As shown in Figure 4, in both cell lines FATP2/ARF6 blocker NAV2729 and FABP4/5/6 inhibitor HTS01037 already impaired FA uptake after 2 h, whereas FABP4/5-specific BMS309403 required longer exposure times, particularly in SKOV3 cells. On the other hand, FABP5- and CD36-targeting agents SB-FI-26 and SSO did not show consistent effects on free FA uptake, respectively. In fact, in SKOV3 cells, SB-FI-26 transiently increased free FA uptake, thereby preventing the determination of an IC_50_ value in these cells (Figure 4).

### 3.3. Inhibitors of Lipid Handling Proteins Diminish the Uptake of Exogenous LDL into OC Cells

Another informative parameter for an on-target effect of the applied metabolic inhibitors would be the rate of cellular uptake of exogenous LDL. LDL is also a major source of FA. It contains phospholipids, cholesterol, and triglycerides. After binding LDL to cognate membrane-anchored LDLR, lipid handling enzymes located in the cell interior disintegrate the constituents and provide free FA as building blocks for remodeling. Heparin, a naturally occurring glycosaminoglycan, forms chelates with LDL and abrogates LDL binding to the cell membranes (Appendix A). Thus, heparin, unlike the synthetic inhibitors, does not directly interfere with any lipid handling protein. Among the synthetic blocking compounds, FATP2/ARF6-specific NAV2729 and FABP4/5-targeting BMS309403 were highly effective. Both blocked LDL import already one hour after drug addition, while HTS01037, SB-FI-26, and SSO were less active (Figure 5). Interestingly, the drug-mediated reduction of cell import was generally more pronounced for LDL than for free FA (Figure 4) and was accompanied by drug-dependent downregulation of LDLR protein levels (Figure 2) and of cell growth (Figure 1). Altogether, pharmacological inhibition of the lipid handling proteins FABP4, FABP5, FABP6, and FATP2, as well as of ARF6 interferes not only with cell proliferation, but also with LDLR expression and with import of FA and particularly of LDL in OC cells.

### 3.4. Inhibitors of Lipid Handling Proteins Do Not Affect the De Novo Synthesis of FA in OC Cells

To get more insight into the effects of the employed lipid uptake and transport inhibitors on the total lipid supply of the cells, we investigated whether these drugs would also affect endogenous FA synthesis. It is well known that during the assembly of endogenous FAs from acetyl-CoA and malonyl-CoA, large amounts of NADPH + H^+^ are required for elongation of acyl-CoA chains, leaving much oxidized NADP^+^ behind. Thus, the content of NADPH + H^+^ relative to NADP^+^ decreases markedly when new FAs are synthesized. Therefore, determination of cellular concentrations of NADPH and its oxidized form NADP^+^ can be used as a surrogate marker for endogenous FA synthesis activity [45]. As shown in Appendix A, none of the agents significantly changed the ratio between NADPH and NADP^+^, indicating that the applied lipid import and transport inhibitors did not affect the de novo synthesis of FA.

### 3.5. Inhibitors of Lipid Handling Proteins Impair Cell Cycle Distribution in OC Cells

For a closer analysis of the drug-induced antiproliferative mechanisms in drug-sensitive OC cells, A2780 cells were exposed for 24 h to NAV2729, BMS309403, HTS01037, or SB-FI-26 and then subjected to cell cycle analysis using flow cytometry detection of PI-stained cells. Data given in Figure 6 reveal that NAV2729 induced a shift from G0/G1 to S and BMS309403 and HTS01037 caused a G2/M block, while SB-FI-26 only slightly decreased G0/G1 and elevated S. The histograms and dot blots from one of three separate experiments for cell cycle analysis by flow cytometry are given in Appendix A.

### 3.6. Inhibitors of Lipid Handling Proteins Promote Apoptosis in OC Cells

Finally, induction of apoptosis, a crucial feature of cancer drugs, was examined in the drug-sensitive OC cells. Activation of programmed cell death was therefore determined in A2780 cells. Data from annexin V/PI labeling and immunofluorescent staining of active caspase-3 followed by flow cytometry are summarized in Figure 7A,B, respectively. They show that a 48-h period of treatment was sufficient to induce apoptosis by each drug in a dose-dependent manner, albeit with different efficiency. Cells were again more sensitive to FATP2/ARF6 inhibitor NAV2729 than to the FABP antagonists BMS309403, HTS01037, and SB-FI-26. Individual dot blots from one of three separate experiments are given in Appendix A, respectively.

### 3.7. Inhibitors of Lipid Handling Proteins Affect Cell Stress- and Metabolism-Pathways in OC Cells

For the assessment of drug-mediated modulation of the expression of gene families and pathways, RNA-seq analyses of whole transcriptomes of A2780 and SKOV3 cells at 4 h and 48 h of drug treatment were performed. Data were analyzed using GO term enrichment and KEGG pathway analysis. As demonstrated in Appendix A, the used inhibitory drugs exerted specific effects on pathways that regulate intracellular cell stress, cell death, cell division, and lipid metabolism in A2780 and SKOV3 cells after acute (4 h) and long-term (48 h) treatment. Overall, NAV2729, HTS01037, and SB-FI-26 affected a greater number of pathways than BMS309403 and SSO.

## 4. Discussion

OC cells not only express high levels of FASN, the rate-limiting enzyme in FA synthesis [12,13,44], and show hyperactive endogenous lipogenesis [15,16,17,18], but they also contain numerous lipid receptors and transport proteins, such as FAT/CD36, LDLR, FABPs, and FATP2/SLC27A2, enabling the cells to vigorously import lipids from extracellular compartments such as ascites [14,46,47,48,49]. Membrane-bound FA channel protein FAT/CD36 and receptor protein LDLR are major entrance ports for lipids. They facilitate transmembrane passage and mediate intracellular trafficking via FABPs and endosomes [50,51]. FAT/CD36 and LDLR are both overexpressed in OC and support aggressive growth patterns and metastasis [20,52,53,54]. Moreover, FABP4 and FABP5—known to be regularly expressed during adipocyte differentiation—have been found at high levels in cancers of the prostate, breast, and ovaries, where they promote malignant progression and metastasis and worsen prognosis [14,55,56,57,58,59]. Strikingly, FABP4 has been shown to directly promote OC metastasis [60] by facilitating the transcellular delivery of FA from omental host adipocytes to the OC cells [4]. Stromal fat cells of the omentum therefore drive metastatic growth of OC cells [4,61,62]. Overall, there is increasing evidence that FAs and lipids are essential not only for proliferation and growth, but also for metastasis of OC and other tumors. Accordingly, targeting FABP4 with the FABP-specific inhibitor BMS309403 significantly reduced cancer cell invasiveness and tumor burden in prostate and ovarian cancer models [21,55]. No severe side effects, such as weight loss or reduced food intake, were observed upon BMS309403 treatment. Therefore, this compound could be used as a promising lead for further cancer drug development [22]. This preliminary evidence prompted us to hypothesize that OC cells are highly sensitive to pharmacological blockade of pathways that regulate the transport of lipids, which may represent a promising strategy to prevent expansion and progression of OC. To this end, we exposed OC cells to inhibitors of the lipid handling proteins FAT/CD36 (SSO), FABP4, FABP5, FABP6 (BMS309403, HTS01037, and SB-FI-26), and FATP2/SLC27A2 (NAV2729), and we determined the on-target efficacy by measuring the reduction of the cellular import of exogenous fluorescence-labeled free FA and LDL into the OC cells. We observed that exposure to the inhibitors downregulated uptake of both free FA and LDL in a compound-, dose-, and time-dependent manner, which correlated with a significant loss of cell proliferation along with induction of cell cycle arrest and apoptosis. There was also a difference in the drug sensitivity between both cell lines. SKOV3 is a multidrug-resistant OC cell line and much more resilient to cell growth inhibition than A2780 [63]. This different behavior may be caused, at least in part, by a different p53 status. A2780 cells express wild-type p53, whereas SKOV3 cells are p53 null [64,65]. Accordingly, the IC_50_ for growth inhibition with SB-FI-26 was more than twice as high in SKOV3 cells compared to A2780, and FA uptake was even increased with a maximum 24 h after drug addition and was decoupled from the concomitant decrease in LDL uptake. The data thus suggest that SKOV3 cells have alternative pathways for free FA uptake that are not available to A2780 cells to compensate for depletion of LDL, which was more pronounced in SKOV3 than in A2780. However, the mechanisms of these metabolic bypass routes and the drug sensitivity of LDL uptake in SKOV3 relative to A2780 cells have yet to be elucidated in detail. Drug-mediated alterations were most prominent after exposure to the FATP2/ARF6 inhibitor NAV2729. According to Grossmann et al. [66], this agent revealed pronounced activity against melanoma, but did not show appreciable in vivo toxicity. This is in line with work from others, who showed that overexpression of lipid handling proteins including FABPs, FAT/CD36, and FATP2 promotes OC cell growth [67,68,69]. Interestingly, although the inhibitors had no appreciable effect on the expression of their cognate targets (FATP2, ARF6, FABPs, CD36), they effectively downregulated LDLR levels, with FATP2/ARF6-specific NAV2729, FABP4/5/6-interacting HTS01037, and FABP4/5-binding BMS309403 showing greater effects than FABP4-blocking SB-FI-26, while FAT/CD36-specific SSO slightly increased the level of LDLR. Downregulation of LDLR may be partially caused by impaired polypeptide elongation and/or accelerated degradation of misfolded proteins due to drug-induced endoplasmic reticulum (ER) stress. Although direct binding of compounds to target proteins may promote molecular decay, the data do not yet suggest that the drugs promoted degradation of their primary high affinity targets. Altogether, drug-mediated downregulation of LDLR was accompanied by a marked decrease in LDL uptake and a modest reduction in FA uptake. LDL import was generally more sensitive than FA import. Overall, this led to decreased cell growth activity. For the FABP inhibitors, a weak trend towards greater inhibitory effects with increased number of FABP targets was observed in the cell growth and FA uptake assays (cf. Figure 1, Figure 2, Figure 3 and Figure 4). Our data demonstrate that the growth of OC strongly depends on exogenous lipid supply. Overexpression of FASN leading to hyperactive endogenous FA synthesis [11,14,15,17,18,44,70,71] is therefore supplemented in OC by an additional process that facilitates the import of exogenous lipids from ascites and omental adipocytes to sustain the increased need for nutrients. Capitalizing on both strategies—autonomous lipogenesis and heteronomous FA/lipid import—gives OC cells a high degree of flexibility. Depending on the conditions, they can pursue de novo fatty acid synthesis, which is a highly energy-demanding reductive anabolic process for regeneration of lipids from non-lipid sources (e.g., glycolysis, glutaminolysis), and/or they incorporate exogenous lipids, which requires much less bioenergy, but depends on extracellular supply of dietary fat. This dual strategy serves best to meet the demand for cellular lipids under harsh growth conditions and to maintain intracellular pH and cell survival [52,72]. Over time, cancer cells have developed a metabolic repertoire to withstand and grow under unfavorable growth conditions [73]. Collectively, our data emphasize the importance of the metabolic requirements of rapidly growing malignant cells and support the idea that OC cells are highly reliant on their microenvironment and particularly on the availability of FA-containing nutrients. Previous findings from us and others [12,70] support the notion that de novo FA synthesis and its key regulator FASN are upregulated in OC cells, and, on the other hand, as shown here, nutrient import pathways mediated by lipid handling proteins are accelerated as well, in order to satisfy the need for biomaterial and energy for cell proliferation and cancer progression. Lipid and/or energy metabolism play crucial roles in carcinogenesis causing characteristic alterations in the metabolic balance. Novel therapeutic approaches that capitalize on these metabolic peculiarities may become very promising anticancer strategies. In the present study we were able to demonstrate (1) that growth and survival of OC cells depends on the orthological function of the cellular lipid handling system and (2) that suppression of these functions by molecular targeted drugs abrogates OC cell growth through a number of distinct mechanisms that include cessation of extracellular FA/lipid supply, discontinuation of cell cycle progression, and activation of apoptosis.

## 5. Conclusions

Preclinical evidence indicates that OC belongs to a group of malignant diseases that is highly sensitive to lipid depletion. Therefore, specific drugs that target this metabolic dependency and block endogenous lipid biosynthesis and/or lipid uptake should be developed for clinical use against advanced OC.

## Figures and Tables

**Figure 1 cancers-14-06004-f001:**
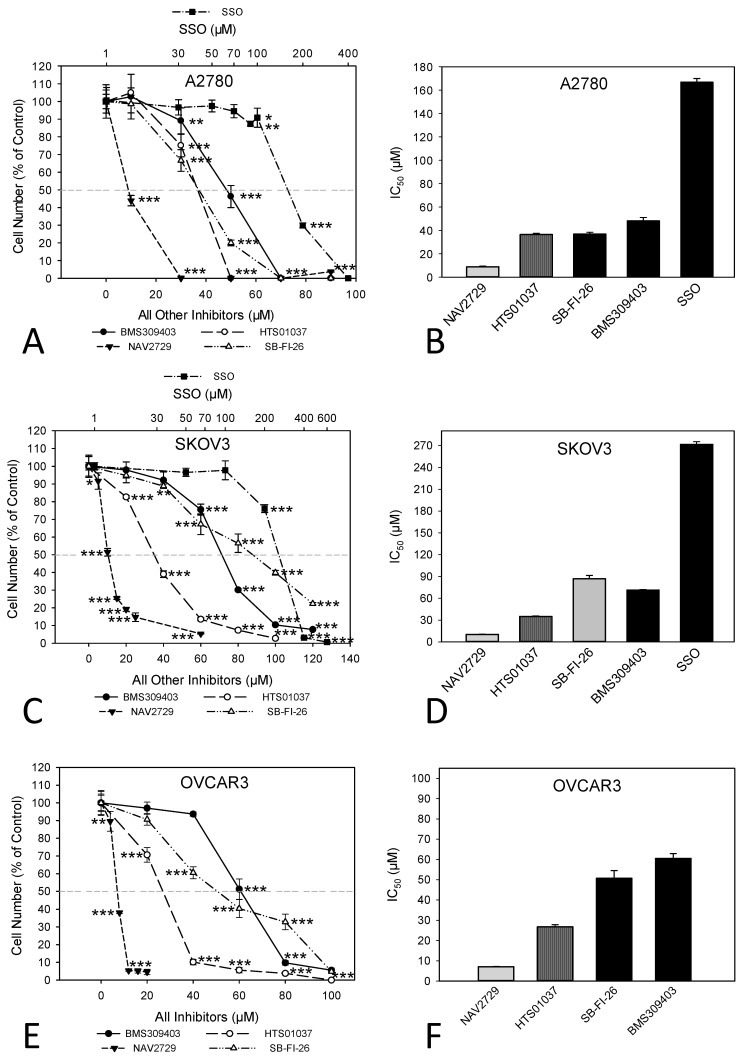
Inhibitors of lipid handling proteins block the growth of OC cells in a dose-dependent manner. Exposure of A2780 (**A**,**B**), SKOV3 (**C**,**D**), or OVCAR3 (**E**,**F**) cells for 72 h to increasing concentrations of small molecule inhibitors of FABPs (BMS309403, HTS01037, SB-FI-26), FATP2, and ARF6 (NAV2729), as well as of FAT/CD36 (SSO), leads to a drug-specific reduction of cell numbers (**A**,**C**,**E**). Note that the concentrations of SSO are plotted on a logarithmic scale at the top of the graphs (**A**,**C**), while the concentrations of all other inhibitors are plotted on a linear scale at the bottom. The IC_50_ values for drug-mediated growth inhibition (**B**,**D**,**F**) vary over a wide dose range with NAV2729 being the strongest inhibitor, followed by HTS01037, BMS309403, and SB-FI-26, and SSO representing a rather inefficient growth inhibitor. Formazan dye assay was used. Means ± SD of three separate experiments with replicates. ANOVA followed by Scheffe test, *p* < 0.05 (*), *p* < 0.01 (**), and *p* < 0.001 (***) relative to solvent-treated cells (0 µM).

**Figure 2 cancers-14-06004-f002:**
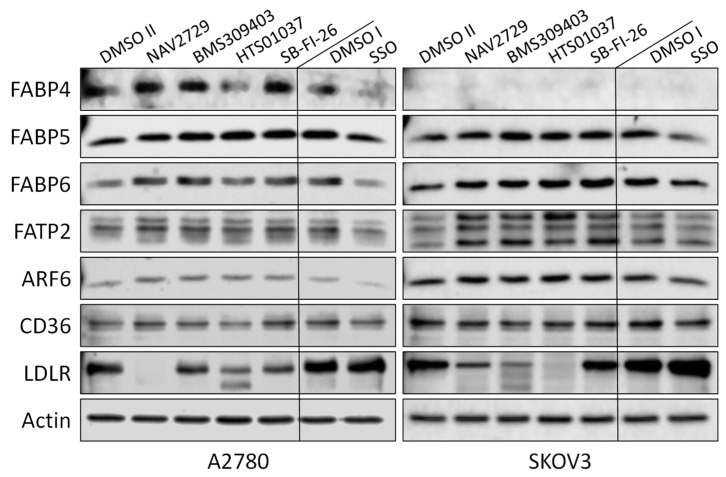
OC cells express a variety of lipid handling proteins. Western blot analysis demonstrates the baseline expression and the effect of pharmacological inhibition of these lipid handling proteins on the expression of the corresponding target proteins FABP4, FABP5, FABP6, FATP2, ARF6, and CD36, as well as of LDLR in A2780 and SKOV3 cell lines. Monolayer cultures were treated for 48 h with drug concentrations close to the IC_50_ values for inhibition of cell growth and uptake of exogenous FA as shown in Figure 1 and Figure 3, respectively (A2780: 7.5 µM NAV2729, 30 µM BMS309403, 22.5 µM HTS01037, 30 µM SB-FI-26, and 100 µM SSO; SKOV3: 10 µM NAV2729, 80 µM BMS309403, 80 µM HTS01037, 80 µM SB-FI-26, and 200 µM SSO) before whole-cell lysates were subjected to SDS-PAGE, immunoblotting, and enhanced chemiluminescence as described in the Materials and Methods Section. Control II was 0.2% (*v*/*v*) DMSO for SSO and Control I was 0.06% (*v*/*v*) DMSO for all other drugs. Actin was used as a loading control. Representative data from one of two separate experiments are shown.

**Figure 3 cancers-14-06004-f003:**
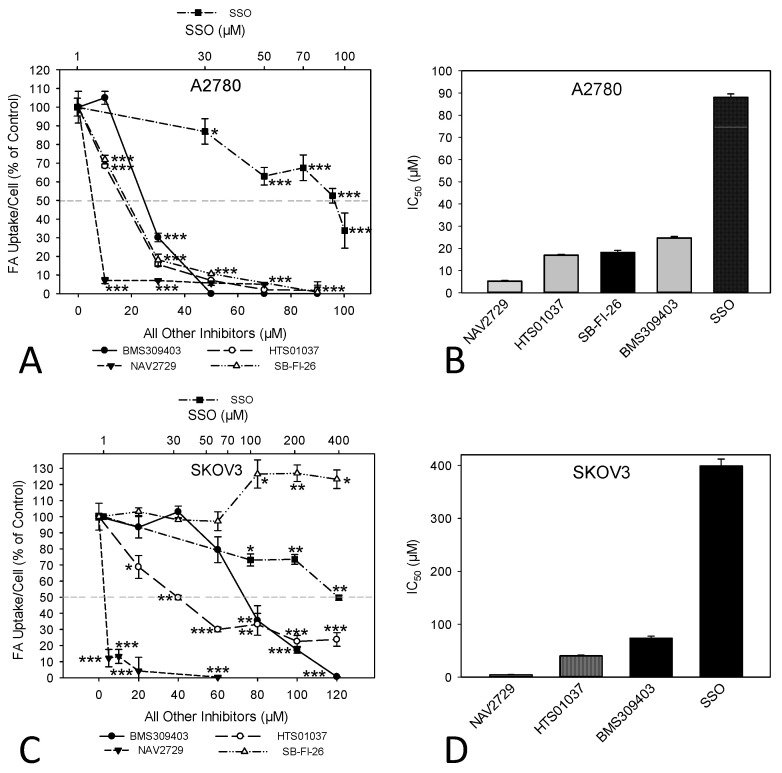
The dose-dependent effect of pharmacological inhibition of lipid handling proteins on the uptake of exogenously added free FAs in A2780 (**A**,**B**) or SKOV3 (**C**,**D**) OC cells. Cells were treated for 48 h with increasing doses of the small molecular inhibitors BMS309403, HTS01037, NAV2729, SB-FI-26, or SSO before FA uptake was ascertained by the QBT™ Fatty Acid Uptake Kit as specified in the Materials and Methods Section (**A**,**C**). Note that the concentrations of SSO are plotted on a logarithmic scale at the top of the graphs (**A**,**C**), while the concentrations of all other inhibitors are plotted on a linear scale at the bottom. Obtained dose–response curves were used to determine the drug-specific IC_50_ values and to find optimal drug concentrations for subsequent time course treatments (**B**,**D**). Fluorescent signals were normalized to the optical density values that were concurrently determined by formazan dye cell proliferation assays as an indirect readout for cell numbers and to the direct cell counts determined visually in a hemocytometer. Normalized data were expressed in % of solvent (0.1% DMSO)-treated controls. Means ± SD of three separate assays with replicates. ANOVA followed by Scheffe test, *p* < 0.05 (*), *p* < 0.01 (**), and *p* < 0.001 (***) relative to solvent-treated cells.

**Figure 4 cancers-14-06004-f004:**
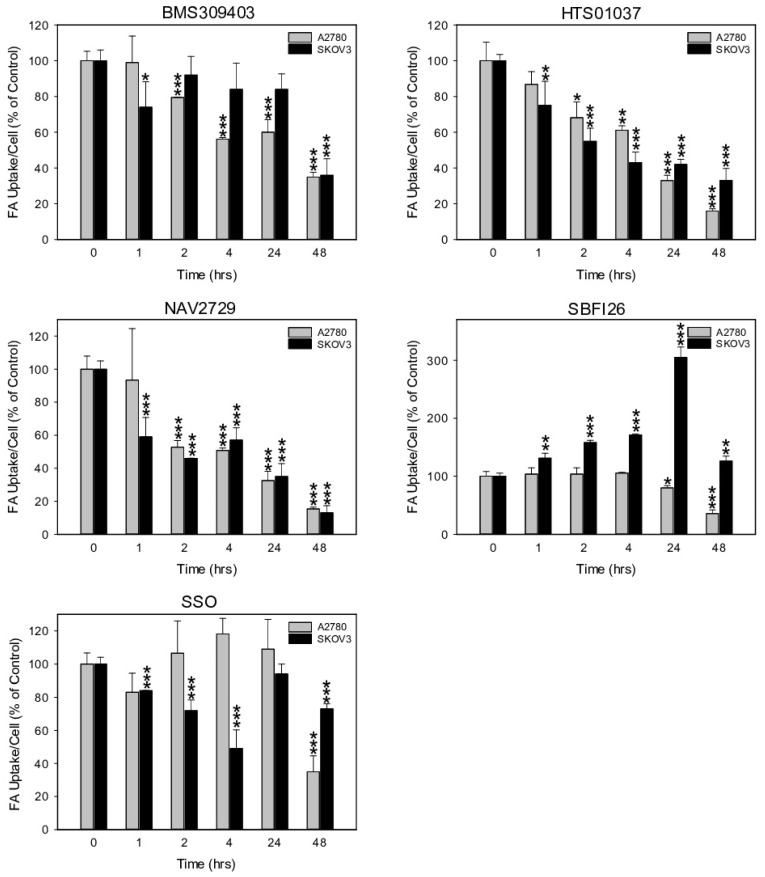
The time-dependent effect of pharmacological inhibition of lipid handling proteins on the uptake of exogenously supplied FAs in A2780 and SKOV3 OC cells. Drug concentrations close to the IC_50_ values for inhibition of cell growth and uptake of exogenous FA (A2780: 10 µM NAV2729, 30 µM BMS309403, 30 µM HTS01037, 30 µM SB-FI-26, and 100 µM SSO; SKOV3: 10 µM NAV2729, 80 µM BMS309403, 80 µM HTS01037, 80 µM SB-FI-26, and 200 µM SSO) were applied for time course experiments. Fluorescence signals were normalized to cell numbers that were concurrently determined by formazan dye cell proliferation assays and hemocytometer counting, and were expressed in % of solvent(0.1% DMSO)-treated control cells. Means ± SD of three separate assays with replicates. ANOVA followed by Scheffe test, *p* < 0.05 (*), *p* < 0.01 (**), and *p* < 0.001 (***) relative to solvent-treated cells.

**Figure 5 cancers-14-06004-f005:**
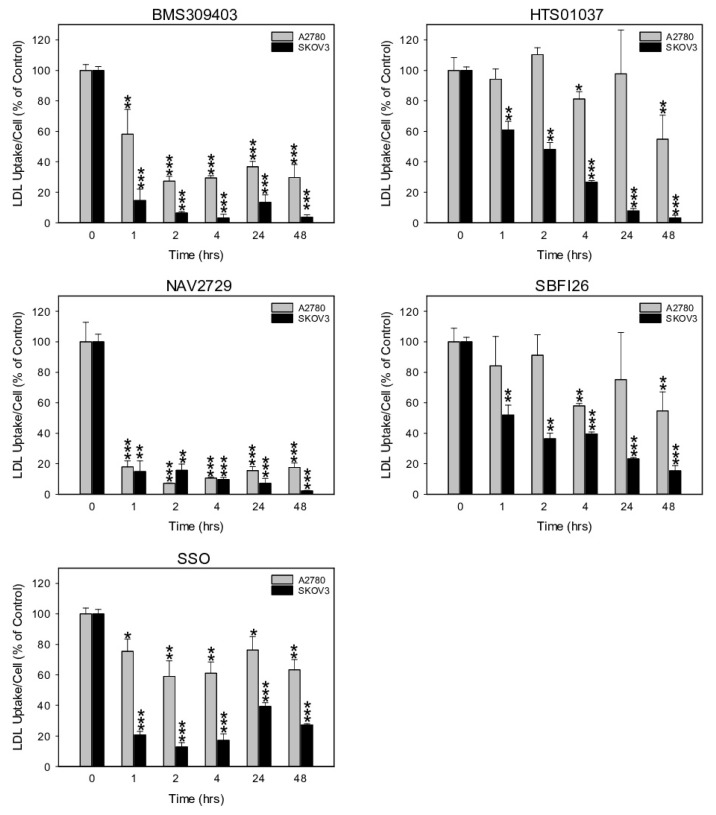
The time-dependent effect of pharmacological inhibition of lipid handling proteins on the uptake of exogenously supplied LDLs in A2780 and SKOV3 OC cells. Cells were exposed for various times to the same doses of the small molecule inhibitors BMS309403, HTS01037, NAV2729, SB-FI-26, or SSO as in Figure 4 before LDL uptake was measured with the pHrodo™ Red LDL Uptake Kit according to the procedure given in the Materials and Methods Section. Fluorescent signals were normalized to the optical density values that were concurrently determined by formazan dye cell proliferation assays as an indirect readout for cell numbers and to the direct cell counts determined visually in a hemocytometer. Normalized data were expressed in % of solvent (0.1% DMSO)-treated controls. Means ± SD of three separate assays with replicates. ANOVA followed by Scheffe test, *p* < 0.05 (*), *p* < 0.01 (**), and *p* < 0.001 (***) relative to solvent-treated cells.

**Figure 6 cancers-14-06004-f006:**
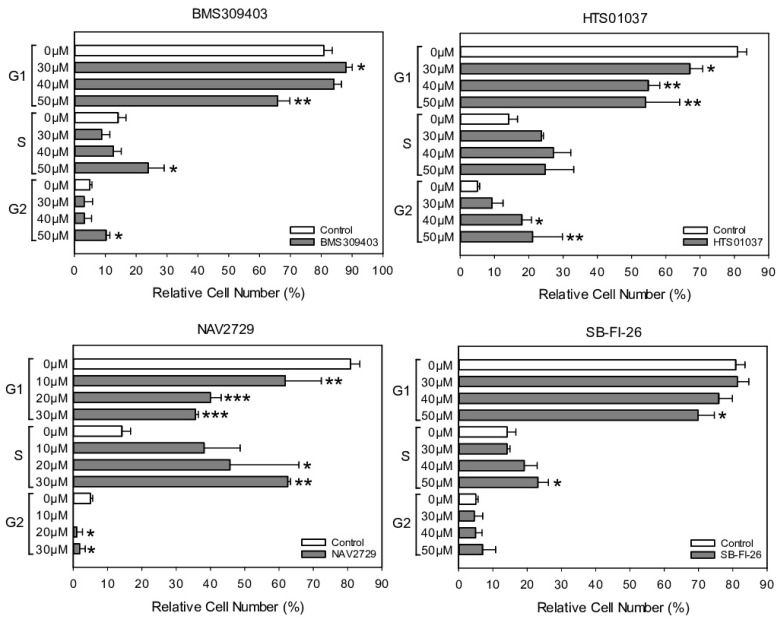
The effect of pharmacological inhibition of lipid handling proteins on the cell cycle distribution of OC cells. A2780 cells were incubated in control medium (0 µM, white bars) or in medium containing various concentrations (gray bars) of small molecule inhibitors of cellular lipid handling proteins at 37 °C for 48 h. Thereafter, cell cycle distribution was analyzed by flow cytometry. Results show relative cell numbers in percent of total and represent the means ± SD of three independent experiments. The level of significance was determined by ANOVA followed by Scheffe test, *p* < 0.05 (*), *p* < 0.01 (**), and *p* < 0.001 (***) compared to control (0 µM, white bars).

**Figure 7 cancers-14-06004-f007:**
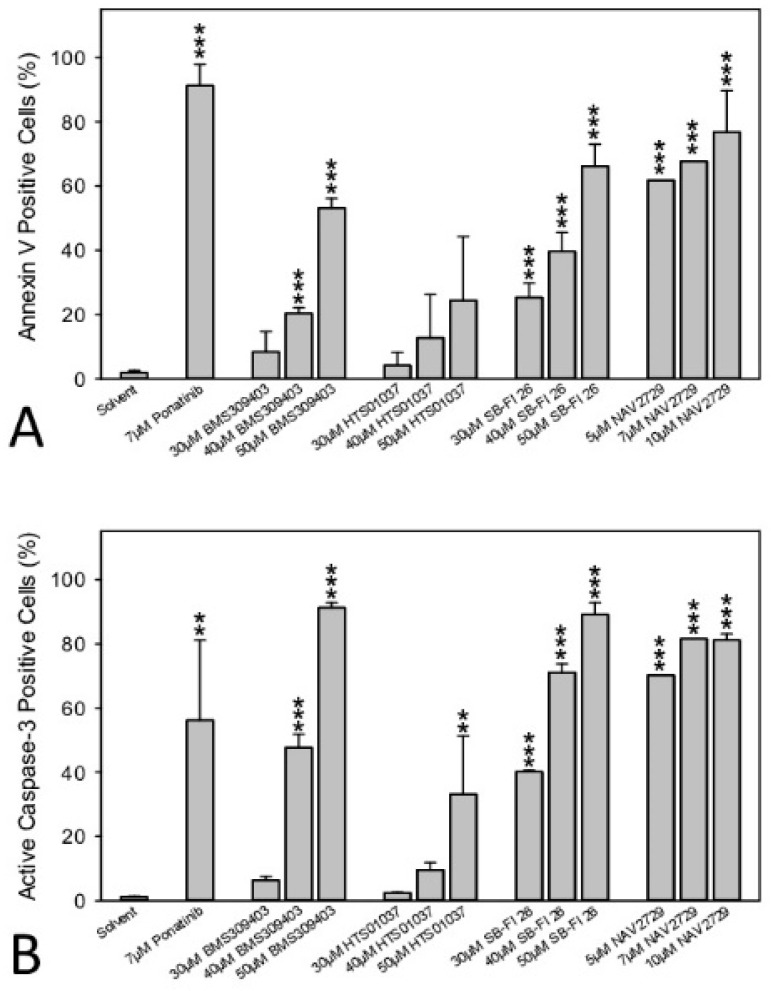
The effect of pharmacological inhibition of lipid handling proteins on apoptosis of OC cells. A2780 cells were incubated in medium containing solvent (negative control), 7 µM of the multi-kinase inhibitor ponatinib (positive control) [39], or various concentrations of small molecule inhibitors of cellular lipid handling proteins at 37 °C for 48 h. Then, cells were examined by flow cytometry to determine the percentage of early apoptotic, APC annexin V-positive cells (**A**) or later apoptotic, active caspase-3-positive cells (**B**). Results show relative cell numbers in percent of total and represent the means ± SD of three independent experiments. The level of significance was determined by ANOVA followed by Scheffe test, *p* < 0.01 (**), and *p* < 0.001 (***) relative to solvent (negative control).

**Table 1 cancers-14-06004-t001:** Key technical data (names, molecular targets in OC cells, sources, and background information) of the lipid handling protein inhibitors used in this study.

Inhibitor	High Affinity Target(s)	Low Affinity Target(s)	Supplier (Cat#)	Reference
NAV2729 (=Grassofermata, CB5)	FATP2	ARF6	Tocris (#5986)	[29,30]
HTS01037	FABP4	FABP5, FABP6	Focus Biomolecules (#10-1453)	[31]
BMS309403	FABP4	FABP5	Tocris (#5258)	[32]
SB-FI-26	FABP5		CaymanChemical (#14191)	[33]
SSO *	FAT **		CaymanChemical (#11211)	[34]

* Sulfosuccinimidyl oleate (SSO); ** aka CD36.

## Data Availability

The RNA-Seq data presented in this study are available in [https://www.ncbi.nlm.nih.gov/geo/query/acc.cgi?acc=GSE212702 on 6 December 2022].

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
