# Peer review of "Pharmacological Inhibition of Lipid Import and Transport Proteins in Ovarian Cancer"

_cancers, 2022, doi:10.3390/cancers14236004_

Round 1

Reviewer 1 Report (Previous Reviewer 1)

Lemberger et al reported the time- and dose-effect on cell proliferation, FA/LDL uptake, cell cycle, and apoptosis using lipid handling protein inhibitors in ovarian cancer cell lines. However, the reports of lipids synthesis as a targetable vulnerability in ovarian cancer were increased. The current study's novelty insight into lipid-antimetabolic strategies against ovarian cancer falls short, in vivo or combined therapy would be appreciated.

Author Response

Reviewer 2 Report (Previous Reviewer 2)

After quickly reading though, I think it is interesting and can be accepted for publication as the authors have addressed all the reviewer's questions.

Author Response

Reviewer 3 Report (New Reviewer)

The Work described by Lemberger et al addresses the effect of inhibiting lipid transporters on lipid uptake and the subsequent effect on proliferation and survival of Ovarian cancer cell lines.  There are certain issues that need to be addressed prior to its acceptance for publication in Cancers.

1. The small molecule inhibitors used for the study - do the affect the translation of the transporter proteins?  There seems to be an effect on some proteins. Like SSO treatment reducing the expression of FABP4, FABP6, ARF6 in A2780. NAV2729 has completely inhibited the expression of LDLR. 

2. Please provide a better blot for FABP4 for Fig 1 - A2780. As per Fig 1, as there is no FABP4 expression for SKOV3, how is the effect mediated for the reduction in cell number by treatment with FABP4 inhibitors HTS and BMS in Fig2. Is it an off target effect on FABP5? That seems hardly the case as SB FI 26 (inhibitor of FABP5) is not as effective as these in fig 2. 

3. It is quite interesting to see that SKOV3 needed nearly double the concentration of drugs than A2780.  What might be the reason ?

4. While a comparison between cell lines and drugs is meant in Fig 2, experiments with OVCAR cell line seem to miss SSO treatment

5. Please reconsider the conclusion drawn in Lines 255 -257. This is not true based on the data provided in fig.1. Also SSO seems to be down regulating several proteins.

6. Figure legends for Fig2. (lines 265, 298) - the scale is not logarithmic as stated. 

7. Please provide an explanation for the difference in IC50 provided in Fig 2 and 3. Likewise, SB-FI is missing for SKOV3

8. NAV2729 has an off target effect on ARF6, an accepted biomarker of OC. How do the authors ascertain that the robust effect observed for NAV2729 is not through ARF6, but through FATP2?

9. Why is there a greater effect of these drugs on LDL uptake than fatty acids? and how do the authors think correlates with the cell proliferation observed?

10. It is difficult to  arrive at a conclusion with Fig6 and Fig 7 - please provide the histograms and dot plots from flow cytometry experiments in the Suppl. data. 

Author Response

Reviewer 4 Report (New Reviewer)

Lemberger and colleagues’ article shows how small inhibitors of different lipid transporters decreases ovarian cancer (OC) cells viability, by repression FA and LDL uptake to the cells.

This is a very interesting approach, which opens the way to the use of such molecules to treat OC.

The manuscript is overall very well written, although not ready for publication.

I would like to suggest to the authors to further detail some of the methods, namely the references for each antibody used, and the specific dilution, as well as the exact seeding number of cells used for each different cell line. At 2.1 it is referred 10 % FBS and then in 2.2 5 % FCS and in 2.3 5 % FBS. Was this reduction in serum just for WB and cell proliferation, during treatments? Please justify. Also for the formazan dye assay, further detail should be given. Was this a MTT assay? Regarding such procedure, in Figure 2, the y axis states “Cell number”, though such assays are not truly giving a number of cells.

The title of 3.1 does not reflect the results showed in Figure 1. In fact, in this section, the authors should describe the WB, and the effects of each inhibitor, which is different depending on the cell line. Was this a representative WB of how many?

The legend of Figure 1 states the drug concentrations used were closed to the IC50, but it is not clear from where such values come from. If they are from Figure 2… such figure should come first.

Still for Figure 1, it is not convincing that all inhibitors (except SSO) downregulated LDLR. This might be true for NAV (in both cell lines), but not so evident for SB, for instance. In fact, the authors forgot to mention and discuss the doble band of LDLR after HT treatment in A2780 cells, and the increase of LDLR after SSO treatment in SKOV3 (sentence in line 440-1 might not be fully true). Also, didn’t SSO downregulated FABP4, FABP6 and ARF6 in A2780 cells?

Please include the results of specific gene expression for confirming your statement in lines 257-9.

At the legend of Figure 2, the authors refer the induction of a “growth arrest”, however, from that figure (based on that procedure), that is not possible to infer. Cells could either be dying or not dividing.

Globally, it is not clear why the authors, for a certain experiment, use 1 or 2 or the 3 cell lines available, without justifying.

Please provide a better introduction to your goals at each sub-title of results 3.2, 3.3, 3.7.

Please specifically discuss the 24 h timepoint of FFA uptake after SBFI26 treatment, and overall why the increase specific to only one cell line.

Please discuss cell line dependency observed in Figure 5.

Please better describe the results obtained in 3.5 (Fig. S3) and 3.8 (Fig. S4 & S5).

Why was not SSO used to evaluate cell cycle nor apoptosis?

At Discussion, I suggest avoiding a repetition of the Introduction.

Please clarify sentence in line 444-5. Was this based on the current manuscript results?

Round 2

Reviewer 3 Report (New Reviewer)

The authors have addressed the queries satisfactorily. I recommend that the manuscript be accepted in the present form for publication. 

Author Response

We thank the reviewer for the effort and time in reviewing our work and for his or her support.

Reviewer 4 Report (New Reviewer)

The authors have adequately answered to all my concerns and suggestions.

Author Response

We thank the reviewer for the effort and time in reviewing our work and for his or her support.

This manuscript is a resubmission of an earlier submission. The following is a list of the peer review reports and author responses from that submission.

Round 1

Reviewer 1 Report

The authors used the lipid-antimetabolic approach to disrupt the import and transport of exogenous fatty acid and lipids and displayed the effects of these inhibitors on cell proliferation, cell cycle, apoptosis, and lipid uptake in ovarian cancer. However, similar research papers have been published, and the novelty is not enough. 

Reviewer 2 Report

In this study, the authors used one ovarian cancer cell line to test the inhibitory effects of several small molecule inhibitors on several genes of the FA/lipid biosynthesis pathway. Although this study provides some preliminary results of these inhibitors, the results are not enough to demonstrate the effects of these inhibitors and their future use for the treatment of ovarian cancer. More experiments including mechanism studies are needed to support the findings.

  1. The authors mentioned that they used three cell lines in the abstract, but actually only one cell line A2780 was used in the results section. This is not enough to demonstrate the inhibition effect of these inhibitors, at least another cell line and patient primary cells need to be tested to confirm the results. Animal study to test the drug efficacy is strongly recommended.
  2. Different inhibitors were used to inhibit proteins in the FA/lipid biosysnthesis, however, the expression of these proteins were not tested. This needs to be tested to confirm the on target effects of these inhibitors. RNA-Seq study is strongly recommended to test whether other genes were affected by the inhibition of each of these inhibitors besides the targeted genes.
  3. The authors showed that the uptake of FA and LDL under the treatment of the inhibitors was decreased, how about the lipid biosynthesis under the same treatment condition?
  4. Figure1A shows significant difference on cell numbers between SSO at 70UM and solvent control (P<0.001), but the difference looks very small to me from the figure. Please check. Same for BMS309403 at 40UM. For figure 1B, what does the standard deviation stands for in the histogram? Based on my experience with IC50 calculation, you can only get one IC50 value with the replicates that you tested and it is the same for Fig2B. Please clarify. In addition, how did you decide the doses for testing for different inhibitors with some having three doses whiles some having five or 6 doses?
  5. The authors mentioned good correlation between cell survival inhibition and FA uptake on page 3, it would be more convincing to test the correlation between cell counts and FA uptake if same dose was used for both experiments for each inhibitor.